# Field Evaluation of an Automated Pollen Sensor

**DOI:** 10.3390/ijerph19116444

**Published:** 2022-05-25

**Authors:** Chenyang Jiang, Wenhao Wang, Linlin Du, Guanyu Huang, Caitlin McConaghy, Stanley Fineman, Yang Liu

**Affiliations:** 1Department of Biostatistics and Bioinformatics, Rollins School of Public Health, Emory University, Atlanta, GA 30322, USA; chenyang.jiang@emory.edu; 2Gangarosa Department of Environmental Health, Rollins School of Public Health, Emory University, Atlanta, GA 30322, USA; wenhao.wang@emory.edu (W.W.); dulinlin1994@hotmail.com (L.D.); caitlin.mcconaghy@emory.edu (C.M.); 3Department of Environmental and Health Sciences, Spelman College, Atlanta, GA 30314, USA; ghuang@spelman.edu; 4Atlanta Allergy and Asthma Clinic, Department of Pediatrics, Emory University School of Medicine, Marietta, GA 30060, USA; sfineman@atlantaallergy.com

**Keywords:** sensors, pollen monitoring, automation, data analysis, real-time monitoring

## Abstract

**Background:** Seasonal pollen is a common cause of allergic respiratory disease. In the United States, pollen monitoring occurs via manual counting, a method which is both labor-intensive and has a considerable time delay. In this paper, we report the field-testing results of a new, automated, real-time pollen imaging sensor in Atlanta, GA. **Methods:** We first compared the pollen concentrations measured by an automated real-time pollen sensor (APS-300, Pollen Sense LLC) collocated with a Rotorod M40 sampler in 2020 at an allergy clinic in northwest Atlanta. An internal consistency assessment was then conducted with two collocated APS-300 sensors in downtown Atlanta during the 2021 pollen season. We also investigated the spatial heterogeneity of pollen concentrations using the APS-300 measurements. **Results:** Overall, the daily pollen concentrations reported by the APS-300 and the Rotorod M40 sampler with manual counting were strongly correlated (r = 0.85) during the peak pollen season. The APS-300 reported fewer tree pollen taxa, resulting in a slight underestimation of total pollen counts. Both the APS-300 and Rotorod M40 reported *Quercus* (*Oak*) and *Pinus* (*Pine*) as dominant pollen taxa during the peak tree pollen season. Pollen concentrations reported by APS-300 in the summer and fall were less accurate. The daily total and speciated pollen concentrations reported by two collocated APS-300 sensors were highly correlated (r = 0.93–0.99). Pollen concentrations showed substantial spatial and temporal heterogeneity in terms of peak levels at three locations in Atlanta. **Conclusions:** The APS-300 sensor was able to provide internally consistent, real-time pollen concentrations that are strongly correlated with the current gold-standard measurements during the peak pollen season. When compared with manual counting approaches, the fully automated sensor has the significant advantage of being mobile with the ability to provide real-time pollen data. However, the sensor’s weed and grass pollen identification algorithms require further improvement.

## 1. Introduction

Exposure to pollen can trigger respiratory illnesses including allergic rhinitis, hay fever, and asthma [1,2,3,4,5]. In the United States, pollen is one of the most common causes of seasonal allergies. Previous studies have found that short-term pollen exposure can significantly increase the risk of allergic rhinitis and asthma, affecting Americans’ health and quality of life [6,7]. Kitinoja et al., reports that an increase of 10 grains per m³ of short-term pollen exposure could result in a 2% increase in the risk of allergic or asthmatic symptoms and a 7% increase in the risk of upper respiratory infection symptoms [8]. The direct medical cost related to allergic rhinitis caused by pollen exposure in the U.S. was estimated to be around USD 3.4 billion, with the dominant component being the prescription medications [9]. In addition, some literature suggest that the health risk of pollen exposure may vary with socioeconomic status and race/ethnicity. For example, Blackwell et al., found that non-Hispanic black adults had 1.13 times higher asthma rates than Hispanic adults [3]. Moreover, allergic response to pollen could be higher among African American children with lower socioeconomic status [10,11].

Pollen-producing plants can be grouped into three categories: trees (e.g., oak, pine, and birch), grasses (e.g., ryegrass and timothy), and weeds (e.g., ragweed, nettle, mugwort, goosefoot, and sorrel). The spatiotemporal patterns of pollen, including start and peak dates, season length, and spatial distribution, are affected by the mixture of pollen taxa and local meteorology [12,13]. Although pollen allergies are common in the clinical setting, the scarcity of pollen data in the U.S. presents a major challenge in advancing our understanding of the temporal and spatial patterns of pollen and the risks of seasonal allergies at the local level. To date, there are only 74 pollen counting stations certified by the National Allergy Bureau (NAB) in the U.S. (https://www.aaaai.org/ accessed on 22 March 2022). These stations have employed a manual counting technique, considered the gold standard for decades. A certified technician is required to identify and count pollen granules collected in the previous 24 h under a microscope. This technique and its associated high labor costs severely limit the spatial coverage and timeliness of pollen data in the U.S. [13].

Automated real-time pollen monitoring offers a promising alternative to manual counting and can improve the spatiotemporal coverage of pollen data. Several methods have been discussed for automated pollen counting, including imaging recognition [14], fluorescence measurements [15,16], biomolecular analysis [17], and laser optics [18]. These methods have various potential strengths and limitations. Fluorescence measurements could be used to discriminate pollen and fungal spores according to their emission spectra. However, fluorescence measurements can only distinguish fungal spores, tree pollen, and grass pollen. Biomolecular analysis method can identify tree pollen and grass pollen by sequencing the chloroplast DNA [17]. This method only requires generic laboratory and bioinformatics techniques instead of trained personnel. However, only three airborne pollen samples were analyzed, and the quantity and accuracy require further investigation. Neither fluorescence nor biomolecular methods have been validated against gold-standard pollen counting methods. Oteros et al. used automatic online pollen monitors with imaging recognition technology to identify most tree pollen taxa with ~85% of data availability [19]. However, their pollen system could only identify 11 tree pollen taxa, and misidentified Betula, Alnus, and Fraxinus that can trigger many allergy symptoms. Additionally, the daily pollen concentration reported by their pollen machine did not show a strong correlation (r = 0.53–0.55) with the gold standard [14].

In this study, we evaluated the field performance of a new commercially available automated pollen sensor, APS-300, developed by Pollen Sense LLC in Atlanta, GA. We compared total and speciated daily pollen concentrations reported by the APS-300 in 2020 with a collocated certified pollen monitor (Rotorod) with manual counting. We also evaluated the internal consistency of APS-300 measurements from two collocated sensors during the peak pollen season in 2021. We further investigated the spatial heterogeneity of pollen concentrations and dominant taxa from three APS-300 sensors deployed in areas with different degrees of urbanization. Finally, we explored the diurnal variability of pollen levels in Atlanta by taking advantage of the real-time pollen counting capabilities of the APS-300.

## 2. Study Design and Statistical Methods

### 2.1. APS-300 Pollen Sense Sampler

Developed by Pollen Sense LLC, the APS-300 is a fully automated pollen imaging sensor that collects and images pollen and airborne particles down to less than 5 μm, in real-time (data reporting delay in <1 min). The APS-300 collects ambient air by an airflow system at a constant flow rate. The particles in the collected air adhered to the rotating tape medium, where a proprietary form of optical surface microscopy is performed. The collection service performs complex proprietary algorithms involving advancing, focusing, and lighting in order to obtain maximal information about each particle. The processing service uses proprietary algorithms to compose these particle images into a single frame. The imaged particles are classified into pollen taxa by neural network algorithms, and the resulting pollen count of each pollen taxon is converted into a daily concentration of pollen granules. The resulting detection and identification data, as well as the composed frame image are then uploaded to the cloud by an uploader service via Wi-Fi or Ethernet. In the event of internet disruption, data is cached until the connection is restored. Given the portable size (slightly bigger than a shoe box) and weight (~5 kg) as well as a simple installation and setup process, the APS-300 can be relocated easily. It also offers the potential for spatially resolved pollen monitoring at the local level. There are currently 25 pollen taxa in the Pollen Sense image library. Currently, it cannot identify the families of Poaceae and Cupressaceae, nor can it distinguish between the genera in these families.

### 2.2. Study Period and Sites

Our field evaluation included two phases. Phase 1 spanned from 1 March to 8 December 2020. During this period, we assessed the accuracy of the APS-300 measurements and the spatial contrast of pollen levels in Atlanta. Three APS-300 sensors were deployed in three sites: (1) Marietta, Georgia, (2) Emory University Rollins School of Public Health, and (3) the Southface Institute, representing various degrees of urbanization in Metro Atlanta (Figure 1). The Marietta site is in the parking lot of a medical complex in a suburban area approximately ~30 km northwest of downtown Atlanta where the Atlanta Allergy and Asthma Clinic (AAAC) is located. Here, an APS-300 and a Rotorod sampler are installed on the same platform ~5 m above the ground. The Rotorod sampler is a rotating arm impactor that collects airborne particles on a rapidly spinning polystyrene rod. The leading surface of the rod is coated with an adhesive to retain any impacted particles. After the collection period, the rod is removed and examined in the lab under light microscopy. The volume of sampled air is then calculated, allowing results to be expressed as particles per cubic meter [20,21]. The pollens collected by the Rotorod sampler at the Marietta site are identified and counted under a microscope by technicians certified by the NAB (last certification exam in 2021). Continuous reviews ensure accuracy and minimize variation between technicians. The Rotorod measurements were used as the gold standard to assess the accuracy of the APS-300.

The Emory site is on a partially covered balcony of the 8th floor of the Claudia Nance Rollins Building on Emory University’s main campus. It is in a moderately built urban area ~10 km northeast of downtown Atlanta. The APS-300 was mounted on a tripod approximately 1 m above the floor (~30 m above the ground) to sample well-mixed ambient air and avoid contamination from resuspended floor dust.

The Southface site is on the roof of the 4-story office building of the Southface Institute located in densely built downtown Atlanta. Like the Emory site, the APS-300 was mounted on a tripod approximately 1 m above the floor (~12 m above the ground). Phase 2 of our field evaluation spanned from 5 March to 31 May 2021. We conducted an internal consistency test with two collocated APS-300 sensors at the Southface site (Figure A1).

### 2.3. Statistical Methods

Three sets of analyses were conducted. First, we compared the pollen measurements of the APS-300 to that of the Rotorod sampler with manual counting. The Rotorod sampler operated by the AAAC in Marietta collects pollen samples for 15 min of each hour for a 24 h period starting in the morning of every weekday. Pollen samples are processed by a certified technician, and the average pollen concentration of the previous 24 h is then reported (i.e., the value reported on Monday morning reflects the average level from Sunday morning to Monday morning). As the Rotorod sampler reports daily pollen levels, the real-time APS-300 measurements were aggregated to 24 h averages to match the collection periods of the Rotorod sampler. Next, we evaluated the internal consistency of the APS-300 measurements from two collocated sensors at the Southface site during the peak pollen season in 2021. Finally, we analyzed total and speciated pollen concentrations measured by APS-300 at all three sites in 2020 to examine the spatiotemporal variability of pollen levels in Atlanta. For these three sets of analyses, total and speciated pollen data were analyzed using the Pearson correlation, time series plots, and bar charts. The tree pollen peak season in Atlanta is usually from late March to early April. However, pollen could reach detectable levels as early as January, and subside by late May. The AAAC reports four levels of tree pollen concentrations to the public, i.e., Low = 0–14, Moderate = 15–80, High = 90–1499, and Extremely High = 1500+. We define the peak season in Atlanta as when pollen concentration reaches High (in fall) and Extremely High (in spring) levels. The spring peak pollen season is from 1 March to 15 April, and the fall peak pollen season is from 1 August to 31 October in 2020.

## 3. Results

### 3.1. Comparison of APS-300 with Rotorod

During Phase 1 of our study, the APS-300 at Marietta obtained valid measurements for 259 days, as compared to 224 days of the Rotorod sampler. The identification level of both APS-300 and Rotorod were genus and family. The APS-300 was able to identify 23 pollen taxa while the Rotorod sampler with manual counting identified 29. The fewer pollen taxa of APS-300 could be due to misclassified or unidentified pollen taxa by its image identification algorithms. The mean (max) total pollen concentration during Phase 1 was 313 (8767) grains/m^3^ estimated by APS-300 as compared to 317 (8863) grains/m^3^ estimated by the Rotorod with manual counting. Figure 2 shows both time series of total pollen concentrations in 2020. Both methods recorded an intensive spring pollen season starting from early March to late April, dominated by tree pollen, and a minor fall pollen season from September to mid-October, dominated by grass and weed pollen. While the manual counting measurements are slightly higher in the peak season, the APS-300 measurements are routinely and significantly higher than the Rotorod during the rest of the year. Due to the observed difference in performance, we analyzed the correlation between APS-300 and manual counting results in the peak season and the rest of the year separately. Total daily concentrations obtained by APS-300 and Rotorod with manual counting showed a strong positive Pearson correlation coefficient of 0.85 (*p* < 0.001; Figure 2A) in the spring tree pollen season. The two sets of measurements are still significantly correlated but with a much weaker Pearson correlation coefficient of 0.47 (*p* < 0.0001, Figure 2B). Table 1 shows the Pearson correlation coefficients for specific taxa. All tree pollen taxa (11 in total) identified by APS-300 except *Ulmus* (*Elm*) are significantly correlated with Rotorod sampler with manual counting. Conversely, among all identified weed pollen taxa (6 in total), only *Ragweed* showed a significant correlation (r = 0.72, *p* < 0.001). APS-300 also reported some pollen taxa that the Rotorod sampler with manual counting did not, such as *Poaceae* and *Chenopodium*.

Figure 3A shows the pollen taxa comparison of weekly total pollen counts at the Marietta site in the peak season. While the Rotorod with manual counting recorded the highest daily pollen count, the highest weekly mean level estimated by APS-300 was slightly higher (the week of 30 March 2020). The top five dominant taxa are individually colored, while the other taxa are summed into one category of ‘others.’ Major contributors at Marietta identified by APS-300 are similar to those identified by the Rotorod with manual counting, but not identical. For example, both methods identified *Quercus* (*oak*) and *Pinus* (*pine*) as the top contributors to pollen levels during this period. However, the APS-300 also identified *Chenopodium*, a large group of annual weedy plants, as a dominant pollen taxon in the peak season, but manual counting did not. It is difficult to tell whether this is a false positive or an improvement. However, some weed pollen taxa, such as pigweed and sheep sorrel, are reported by Rotorod with manual counting but not included in the APS-300 image library. We speculate that there might exist some misclassification between *Chenopodium* and other weed pollen by the APS-300. In addition, some pollen taxa in the image library, such as hazelnut and mugwort, were collected elsewhere, and are probably not sufficiently representative of these pollen particles in Atlanta. As a result, APS-300 was not able to identify them in our study.

### 3.2. Internal Consistency of APS-300 Measurements

Figure 4 shows similar time trends of total pollen concentrations from the two co-located APS-300 sensors at the Southface site. One unit recorded slightly higher overall pollen levels and a 50% greater peak value in early April of 2021. The scatterplot in Figure 4 indicates excellent overall agreement of the two sensors with a Pearson correlation coefficient of 0.99 (*p*-value < 0.001). Both units identified 12 tree pollen taxa and one weed pollen taxon. All pollen taxa (Table 2) measured by the two sensors during the consistency test were strongly correlated with Pearson correlation coefficients greater than 0.93. The scatter plot and simple linear regression in Figure 4 indicated a highly linear relationship between the two sets of measurements in the entire data range.

### 3.3. Spatiotemporal Variation of Pollen Counts

To study the spatiotemporal variation of pollen levels in Atlanta, we analyzed the APS-300 measurements at Marietta, Southface, and Emory in 2020. Across the three sites, the average daily total pollen concentrations ranged from 281 grains/m^3^ at Marietta to 561 grains/m^3^ at Emory. In terms of peak pollen counts, the Marietta site reported the highest level of over 8800 grains/m^3^ while, the Southface site (Southface 1) had the lowest peak level of 5933 grains/m^3^. Figure 5A shows that, during the spring peak pollen season (1 March–15 April), Marietta had the most concentrated peak pollen season and Emory had the longest and most dissipated pollen season, with Southface between the two. Unlike Marietta’s distinct peak, pollen levels at Emory and Southface fluctuated with multiple peaks. During the fall peak pollen season (1 August–31 October), pollen levels at Marietta were lower than Emory and Southface. The daily pollen levels measured at each site are all significantly correlated, and Southface and Marietta have the lowest Pearson correlation coefficient of 0.53, probably because they are the furthest apart and have different vegetation types (Figure 5C–E).

Marietta and Southface sites identified 23 different pollen taxa. On average, 98%, 96%, and 97% of total pollen grains were speciated at Marietta, Southface, and Emory, respectively. Figure 3B shows the weekly dominant pollen taxa at these three sites. Overall, all three sites identified similar dominant pollen taxa including *Quercus* (*oak*), *Pinus* (*pine*), *Betula* (*birch*), *Carya* (*hickory*), and *Chenopodium*. At the weekly level, the dominant pollen taxa varied. For example, during the week of 30 March, *Pinus*, *Quercus*, *Betula*, and *Chenopodium* contributed the most to total pollen levels at Marietta and Southface, while *Carya*, *Betula*, and *Chenopodium* were the dominant taxa at Emory.

Taking advantage of APS-300’s real-time sampling capabilities, we investigated the diurnal variability of pollen levels at our study sites from 24 March to 31 March 2021, which measured the highest pollen levels during our study period. To understand the local meteorological drivers of hourly pollen concentration patterns during this peak week, we extracted the surface level hourly meteorological fields at each site generated by the High-Resolution Rapid Refresh model (HRRR) model [22]. The HRRR model is a numerical atmospheric model that provides hourly weather forecasts at 3 km spatial resolution in the U.S. [23]. The frequency of the meteorological parameters determined the aggregation of real-time APS-300 measurements to the hourly level. We also averaged the hourly pollen concentrations during this week to reduce day-to-day fluctuations due to weather conditions. Figure 6 shows that all three sites followed a similar pattern: hourly pollen levels were the lowest from early morning (4 a.m.) to noon, followed by a sharp increase from noon to approximately 2 p.m. where they remained relatively high until approximately 9 p.m., after which pollen levels began to decrease. We performed site-specific Pearson correlation tests between various meteorological parameters with hourly pollen concentrations (Table A1). Overall, pollen level has a moderate and positive correlation with temperature and boundary layer height, and a negative correlation with relative humidity. The correlation with surface-level wind speed was significant at both Southface and Emory, but was not at Marietta.

## 4. Discussion

The motivation of this study was to evaluate the potential of the APS-300 in investigating how pollen levels and speciation vary in space and time at the urban scale. To this end, we designed our study to focus on the accuracy and internal consistency of the APS-300. We first compared APS-300 with a NAB certified Rotorod sampler to evaluate their agreement of pollen time trend, analyzed the correlation of total and speciated pollen levels, and compared the dominant pollen taxa of these two methods. In our study, the APS-300 demonstrated similar abilities when compared with the Rotorod sampler with manual counting. The correlation coefficient of the daily pollen concentrations measured by the APS-300 and the Rotorod sampler with manual counting was as high as 0.85 in the peak pollen season. The APS-300 was able to process and analyze pollen taxa soon after an image of pollen grains on the sampling tape was captured, which allowed us to analyze pollen trend, report pollen level, and identify up to 23 pollen taxa in near real-time. Both the APS-300 and the Rotorod with manual counting identified *Quercus* (*oak*) and *Pinus* (*pine*) as the dominant pollen taxa during the peak pollen season in Atlanta. The concentrations of major tree pollen taxa measured by the APS-300 are strongly correlated with those of the Rotorod sampler with human counting except *Ulmus* (*Elm*) (Table 1). Rotorod collected mostly fall *elm* pollen; however, the APS-300 only captured some spring *elm* pollen. The concentration of fall *elm* pollen captured by the APS-300 is much lower than the manual counting results, which is the main reason for the insignificant correlation. The weak correlation results of weed and grass pollen indicate that the APS-300 system needs to strengthen its image library of these taxa to better train the pollen identification algorithm. Several previous studies also compared automatic pollen monitors with a Hirst type sampler, another NAB certified sampling method [14,19,24]. For example, Oteros et al. evaluated a BAA500 automatic pollen sensor which also uses imaging recognition technology. They found that, during the pollen season, the correlation coefficient of daily total pollen count between their sensor and a Hirst type sampler was 0.98. The BAA500 automatically identified 16 pollen taxa, 9 of which had a correlation coefficient of over 0.9 with the Hirst sampler measurements [19]. The BAA500 automatic pollen sensor has a higher correlation compared with Hirst-type traps with manual counting; however, the APS-300 is much smaller, cheaper, and easy to operate. In addition, the APS-300 has a shorter delay time (<1 min) compared with the BAA500 (3 h); therefore, we could see hourly pollen concentrations immediately.

The consistency test showed that daily total pollen concentrations measured by two collocated APS-300 devices were highly correlated with a correlation coefficient of 0.98. The simple linear regression analysis suggests that systematic differences between two identical APS-300 sensors can exist but are likely small in relation to the average pollen level (Figure 4). The automatic pollen counting and identification algorithm used by the APS-300 provides particle recognition in size and range of most airborne particles, such as pollen, molds, dust, pollutants, and various other elements that may be organic, naturally occurring, or synthetic. The discrepancies shown in our consistency test could be due to the uncertainties in weed and grass pollen identification, as well as other particle types.

The APS-300 measurements at three sites are significantly correlated, but the correlation grew weaker with distance. This is expected, as weather conditions can help explain the distance-dependent correlation of pollen levels among sampling sites, e.g., similar onset of the pollen season driven by temperature and similar total pollen counts driven by horizontal mixing by wind. A closer examination of the temporal trends showed more noticeable differences in terms of absolute levels, temporal patterns, and dominant taxa. For example, the maximum daily pollen concentrations during Phase 1 ranged from ~6000 grains/m^3^ within the city perimeter (Emory and Southface) to ~8000 grains/m^3^ in the suburb (Marietta). This could be due to different dominant pollen-emitting species at these sites. The spatial contrast of pollen levels within a city has been reported in a handful of studies before. For example, Weinberger observed that total tree pollen influx ranged from 2942 grains/cm^2^ to 17,460 grains/cm^2^ across monitoring sites in New York City [25]. A previous study also suggested that pollen concentration trajectories over an entire pollen season driven by the local conditions were different among different areas of the city [26]. Precipitation has a strong impact on pollen levels. We did not observe any precipitation events during our peak pollen season, therefore were unable to assess its correlation with pollen levels in our study. To test any hypothesis of the drivers of the pollen spatial contrasts will require additional monitoring sites with varying weather conditions and land cover types, which is beyond the scope of the current study. It is our future research direction.

The differences in the temporal trends of pollen levels among our study sites are more substantial. Figure 5A indicates that Marietta had the most concentrated and shortest pollen season, while the other two sites had an earlier start date of the peak pollen season, multiple peak days, and a longer pollen season. This might be due to the differences in quantity of pollen-emitting taxa, land use patterns, and microclimate among these sites. In general, our findings agree with a systematic review which reported that pollen season duration was correlated with its start date, where locations with earlier start dates have longer pollen seasons [13].

Compared with the current gold-standard pollen counting technology, the APS-300 has a significant advantage of providing near real-time pollen concentrations. As we explored the hourly variability of pollen levels in Atlanta, we observed a similar diurnal pattern shared by all three sites during the peak pollen week, i.e., the lowest pollen levels appear around noon, followed by a period of relatively high pollen levels until midnight. While most previous studies addressed the relationship between pollen levels and meteorology factors at the daily to monthly level, few have examined the hourly variation of pollen due to the lack of high-frequency data [27]. For example, Hernández-Ceballos et al. assessed the intra-diurnal variations in airborne oak pollen levels at two sampling stations 40 km apart in Spain. The pollen grains were captured using Hirst-type volumetric spore traps at the bi-hourly level followed by manual counting [28]. Thibaudon et al., aimed to detect the origin of and the weather impact on ragweed pollen in France by collecting bi-hourly pollen grains with Hirst-type traps and manual counting under a microscope [11]. Our correlation analysis suggests that pollen diurnal variability might be driven by temperature, relative humidity, and boundary layer height at the hourly level. These findings are consistent with previous evidence that the increasing temperature and decreasing relative humidity enhance the proportion of dehisced anthers [29]. As far as we know, our study is the first attempt to explore the association between hourly pollen concentration measurements and weather conditions. We speculate that higher temperature and greater boundary layer height are usually associated with stronger convection and vertical mixing, causing larger pollen particles to resuspend and be transported [30]. A possible explanation of the negative association between humidity and pollen concentration is that it has been reported that fresh pollen grains can rupture at high humidity and during precipitation events and change to sub-pollen particles [31] which are more challenging for the APS-300 to count and identify. The effect of wind is harder to explain. One possibility is that the APS-300 is placed much higher above the ground at both Southface and Emory than Marietta. The effect of wind transport might have a stronger impact on pollen levels at these two sites [32]. However, meteorological factors alone are insufficient to fully address the temporal variability of pollen concentrations. Future research that also considers pollen production patterns is warranted.

## 5. Conclusions

Our ultimate goal is to develop a spatially resolved pollen model at the urban scale using pollen measurements from ground monitors, land cover types, and high-resolution meteorology. Such a task requires a denser pollen monitoring network. Our field evaluation of the APS-300 sensor showed statistically significant and strong correlations between its measurements of total and speciated tree pollen when compared with the current gold standard. It also demonstrated strong internal consistency despite the slight systematic differences. We also identified areas in need of future improvement, including the counting and speciation of grass and weed pollen. Our spatial analysis indicated that pollen levels and their temporal trends vary within the city of Atlanta. Therefore, a portable and automatic pollen sensor such as the APS-300 can be a useful tool to better understand the spatiotemporal variability of pollen at a finer spatial scale, which may be linked to community-level disparities in allergic respiratory illnesses. Going forward, we will continue exploring the relationship between pollen levels and various meteorological, land cover, and land use parameters in order to build a comprehensive spatial pollen exposure model which will support the research of pollen-related health effects.

## Figures and Tables

**Figure 1 ijerph-19-06444-f001:**
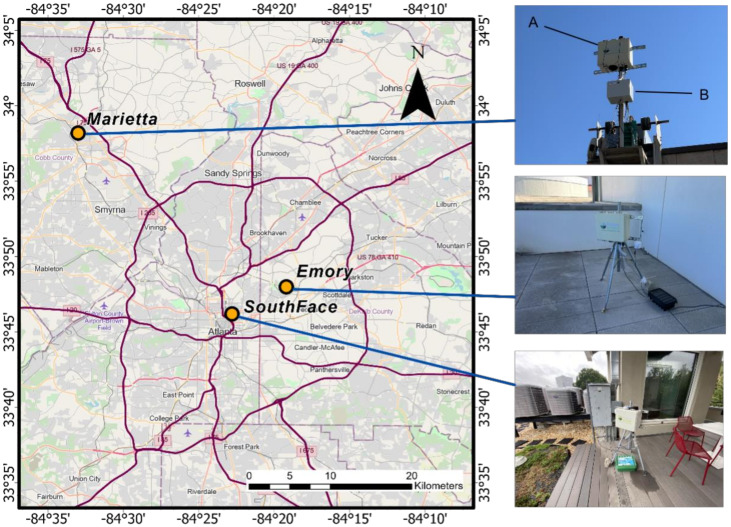
Study domain and sites. More descriptions on map. Device A is the APS-300 while device B is the Rotorod.

**Figure 2 ijerph-19-06444-f002:**
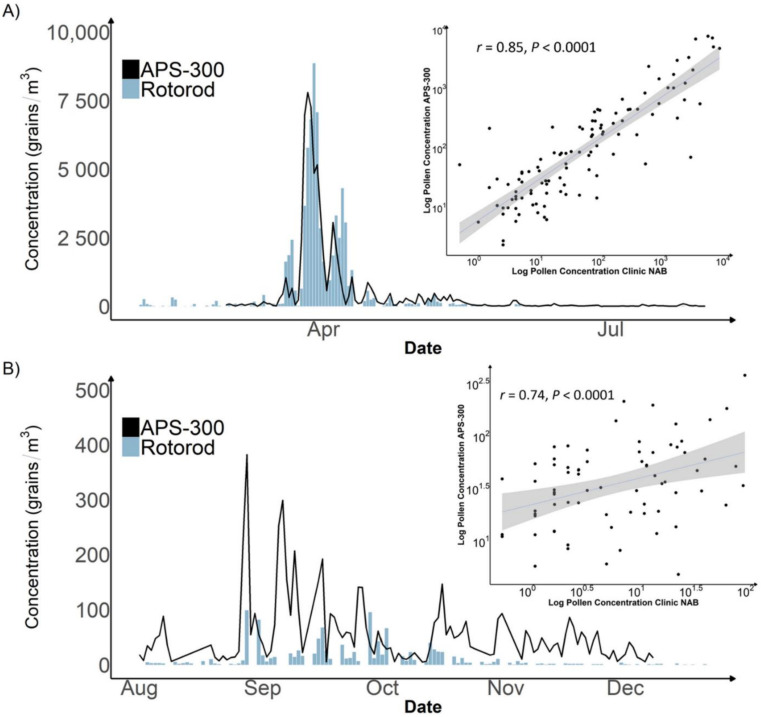
Time Series Plot and Scatter Plot of Daily Pollen Concentration Measured by Rotorod sampler with manual counting vs. APS-300 in Marietta, Atlanta, 2020. Plot (**A**) displays a time series plot of the peak pollen season with a Pearson correlation plot. Plot (**B**) displays a time series plot of the off-peak pollen season with a Pearson correlation plot.

**Figure 3 ijerph-19-06444-f003:**
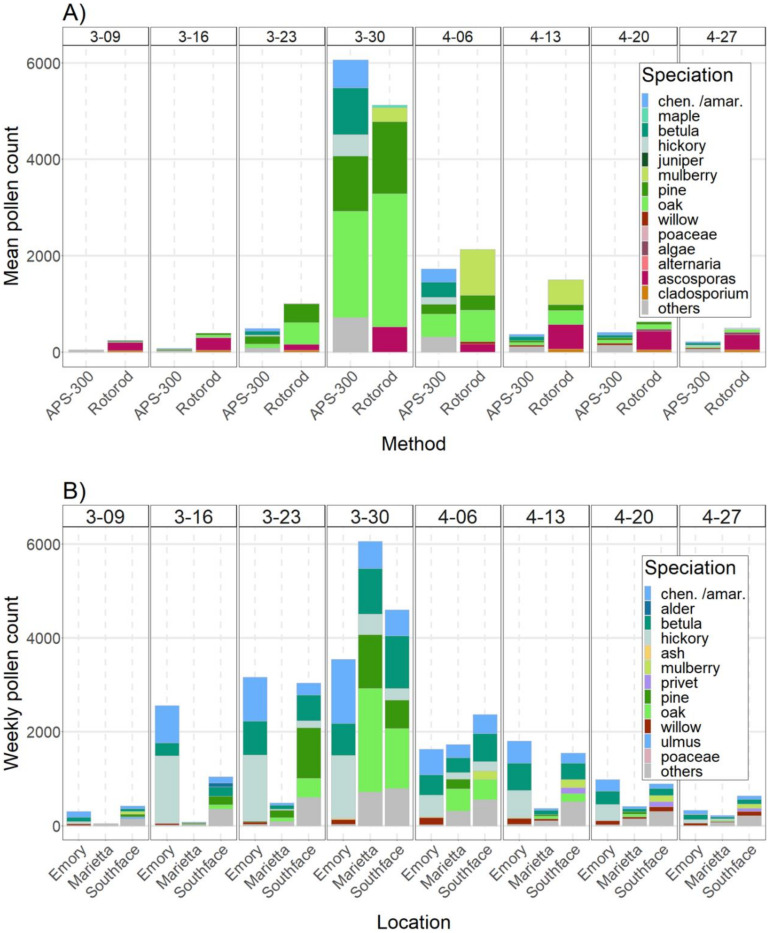
Dominant pollen taxa in the peak season of 2020. Plot (**A**)—comparison of dominant pollen taxa measured by Rotorod sampler with manual counting and APS-300 at Marietta site. Plot (**B**)—comparison of dominant taxa measured by APS-300 in Marietta, Southface, and Emory.

**Figure 4 ijerph-19-06444-f004:**
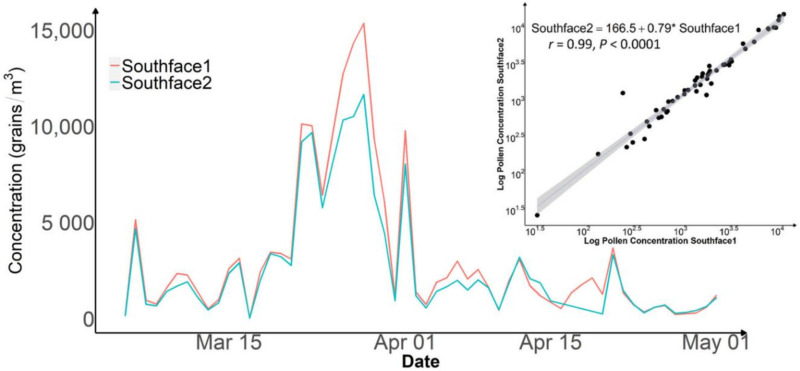
Time series plot and scatter plot of daily pollen concentration measured from the consistency test at the Southface site in 2021. *: multiplication.

**Figure 5 ijerph-19-06444-f005:**
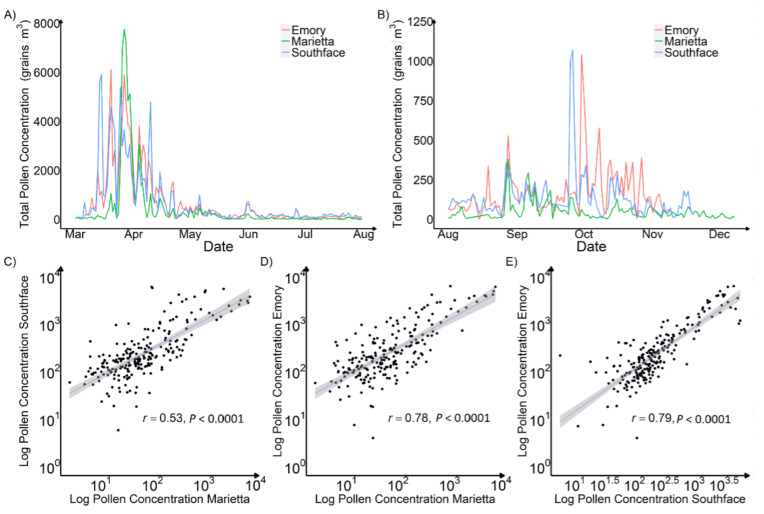
Time Series Plot and Scatter Plot of Daily Pollen Concentration Measured by APS-300 in Marietta, Southface, and Emory in 2020. Plot (**A**) is a spring peak and off-peak pollen season time series plot. Plot (**B**) is a fall peak and off-peak pollen season time series plot. Plot (**C**) is a scatter plot between measurements at Marietta and Southface. Plot (**D**) is a scatter plot between measurements at Marietta and Emory. Plot (**E**) is a scatter plot between measurements at Southface and Emory. The peak pollen season was from late March to mid-April.

**Figure 6 ijerph-19-06444-f006:**
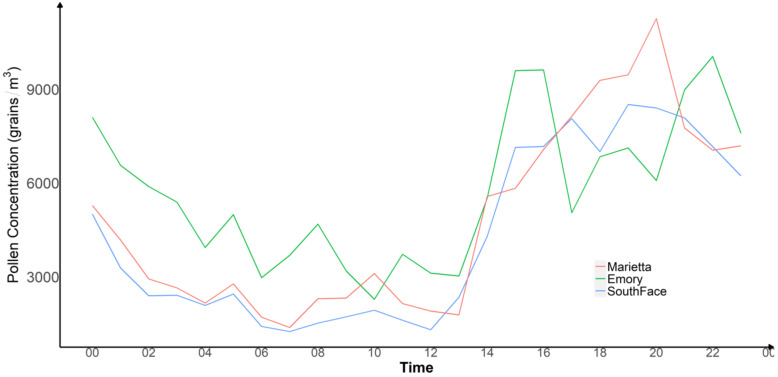
Real-Time Pollen Monitoring of Three Devices from 12 p.m. of 24 March to 12 p.m. of 31 March 2021.

**Table 1 ijerph-19-06444-t001:** Pearson Correlation of Matched Pollen Taxa Concentration measurements (Rotorod vs. APS-300).

Category	Pollen Taxa	Mean (Rotorod)	Mean (APS-300)	Max (Rotorod)	Max (APS-300)	r (*p*-Value)
Tree	*Quercus*	531.5	366.6	4752.7	3390.2	0.85 ***
	*Pinus*	262.6	159.8	2754.9	1452.7	0.85 ***
Tree	*Carya*	6.1	26.3	141.4	529.9	0.77 ***
Tree	*Morus*	60.6	4.8	3095.5	87.6	0.76 ***
Tree	*Fraxinus*	1.7	7.6	41.2	160.6	0.71 ***
Tree	*Betula*	15.4	241.2	136.7	1273.3	0.66 ***
Tree	*Salix*	18.9	41.9	104.1	197.7	0.57 ***
Tree	*Acer*	3.4	1.7	301.5	42.7	0.51 ***
Tree	*Alnus*	0.2	6.4	6.5	113.0	0.38 ***
Tree	*Cupressaceae*	2.6	0.2	36.9	10.3	0.28 **
Tree	*Olea*	0.4	12.8	23.9	236.1	0.28 **
Tree	*Populus*	0.3	2.0	15.4	36.9	0.21 ***
Weed	*Ambrosia/Iva*	2.6	1.9	36.7	49.4	0.72 ***
Tree	*Ulmus*	2.7	1.7	91.7	37.0	0.05 ^#^
Weed	*Asteraceae*	0.1	2.9	3.0	29.5	0.03 ^#^
Weed	*Plantago*	0.4	6.5	9.5	84.8	−0.02 ^#^
Weed	*Urticaceae*	0.2	0.4	5.9	9.4	−0.05 ^#^
Weed	*Rumex*	0.6	0.7	21.7	18.5	−0.08 ^#^
Weed	*Cyperacease*	0.1	1.3	2.2	49.4	−0.11 ^#^
	Overall	350.3	332.1	8863.3	7753.8	0.77 ***

^#^: *p* > 0.05, **: *p* < 0.01, and ***: *p* < 0.001.

**Table 2 ijerph-19-06444-t002:** Pearson Correlation of Matched Pollen Taxa (Southface1 vs. Southface2).

Category	Pollen Taxa	*r* (*p*-Value)
Tree	*Betula*	0.99 ***
Tree	*Alnus*	0.99 ***
Tree	*Fraxinus*	0.99 ***
Tree	*Ulmus*	0.99 ***
Tree	*Carya*	0.99 ***
Tree	*Populus*	0.99 ***
Tree	*Pinus*	0.98 ***
Tree	*Acer*	0.97 ***
Tree	*Morus*	0.97 ***
Tree	*Salix*	0.95 ***
Tree	*Quercus*	0.93 ***
Tree	*Cupressaceae*	0.93 ***
Weed	*Plantago*	0.99 ***

In the period of consistency test (5 March 2021–1 May 2021), there were no observations of Olea, Chenopodium/Amaranthus, Urticaceae, and Poaceae. ***: *p* < 0.001.

## Data Availability

Data may be become available upon reasonable request.

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
