# Peer review of "Field Evaluation of an Automated Pollen Sensor"

_ijerph, 2022, doi:10.3390/ijerph19116444_

Round 1
Reviewer 1 Report
Summary:
The main goal of this study is to field-test a new automated real-time pollen sensor. The paper also does a spatial analysis of pollen concentration using 3 sites, and diurnal analyses of pollen concentration.
I am excited to see this study and think a comparison of the new APS300 sensor to a Rotorod “gold standard” is sorely needed. The strength of this manuscript is in this comparison.
The study also does spatiotemporal analysis using the new sensors. However, this part of the study does not seem well-thought out and could use more substance and rigor. I would like to see (1) background literature, (2) hypotheses or a demonstration of understanding of processes behind spatiotemporal variability and (3) quantitative results of these analyses.
The manuscript is well written and easy to understand.
Specific comments:
L47: Mudarri (2016) paper deals with allergic disease associated with dampness and mold, not pollen.
L94: “urban, suburban, and rural areas”. The manuscript doesn’t seem to focus on the urban, suburban and rural difference. The sites are categorized as “suburban, suburban, and urban areas” (line 117), is this a typo or are you lacking a rural site? Later in the paper, when presenting the results of the 3 sites, the urban/suburban/rural category does not seem to contribute to understanding the difference between the 3 sites. Did you have a hypothesis? Did you expect to see a difference between the sites or not? And why, or why not?
L99: How does the APS300 sensor collect pollen? A more detailed description would be great. L31 mentions that an advantage of this sensor is it is mobile – maybe you could describe what makes it mobile - is it the weight, the ease to set up, etc?
L101-103: How many pollen taxa are in the image library? One of the limitations of humans using traditional microscopy to identify pollen grains is that we can’t distinguish between the genera in the families of Poaceae and Cupressaceae. Is this possible with the APS300? If so, this would be a great improvement. I can’t quite tell from reading the manuscript, but it appears that the APS300 cannot distinguish between genera for those families either. Please make it clear in the text. Could you provide a little more description of how the APS300 collects the pollen and how it images the pollen?
L121: “8th floor” and L124 “4-story” How many meters are these above the ground? I’m also concerned that the Emory site is “on a partially covered balcony”, won’t this affect the number of pollen grains collected by preferentially sampling pollen from the direction of the uncovered area?
L127: How far apart are the two APS300 at Southface? Are they facing the same direction? Does direction matter?
Section 2.2: Please add in the methodology of collection, counting and identification using the Rotorod. Include any error in collection? Is there more than one technician counting and identifying the pollen? If so, how do you account for the differences between technician? Is the Rotorod not “mobile”, like the APS300? If so, what makes it not mobile?
Fig1 : Does the Marietta photo have both the AP300 and Rotorod in the photo? If so, could you make it label them?
L150-151: “The APS-300 was able to identify 23 pollen species while the Rotorod sampler identified 29 species.”
- Could you provide more information on why APS300 identified less pollen taxa? Did the APS300 not have the 6 species it didn’t identify in its image library? Or are they misidentified? Are the misidentified in a consistent or random way?
- The identification level is genus and family, not species (at least using the Rotorod). I don’t know for the APS300.
- Technically the Rotorod doesn’t identify the pollen taxa, a person does. The Rotorod is just the collection device. This reference to Rotorod identifying pollen type is made throughout the manuscript. Please fix all occurrences of this.
L160-162: How did you define “peak season”? How many days were in the peak season vs the rest of the year?
L167: “except Elm are significantly correlated”. I know that Atlanta has fall pollinating Elm trees, whereas many northern NAB station locations have only spring pollinating Elm trees. Do you think this is the reason for this?
Fig 2: It’s not clear what is the blue bars and black line on the figure, please label as APS300 or Rotorod. Fig2B inset cuts off the main figure.
Table 1: The columns listed as “Species (Rotorod)” and “Genus (APS300)” should use consistent naming either scientific names or common names. The “Species” column should not be species, but genera and families. I suggest using the term “taxa” to refer to both genus and family.
L178: “speciated” The pollen taxa aren’t divided by species.
L180: “week 13” what date does this refer to? Please be consistent in labeling times with either dates or weeks with Fig 3.
Fig 3: Please use different colors, it’s very hard to distinguish between the greens.
Table 2: Cupressaceae is not a genus, please rename the column.
Fig 4: The inset has cut off larger figure.
L208-227: This beginning half of this section deals with the spatial variation of pollen counts by looking at the 3 different sites. This is an interesting idea, but I think to include this into the manuscript it needs substantially more work. What do you want to learn from this spatial analysis? Did you expect the sites to have similar pollen concentrations or not? Similar pollen taxa or not? Are the plants in the immediate area the same or different for the 3 sites? Some studies have shown that the height of the sensor affects the pollen concentration (https://doi.org/10.3390/atmos11020145) so I’m not sure what conclusions you can draw between the 3 sites because the heights of the sensors are all different.
L210: I assume you are using the APS300 sensor measurements at Marietta, not the Rotorod? Please be clear.
L211: Which Southface sensor are you using, 1 or 2?
L237: “to reduce high-frequency fluctuations due to weather conditions” Weather conditions have a major effect on the amount of pollen in the atmosphere, so why do you want to reduce the high-frequency fluctuations? I think if your goal is to understand when pollen is being emitted into the atmosphere by plants, then reducing the fluctuations due to weather conditions is a good idea. However, later in this paragraph, you are correlating weather variables with pollen concentration. So, you need to figure out what you want to understand about airborne pollen concentration – is it the diurnal cycle of plant pollen emission or is it how weather affects the pollen concentration?
L237: Is Fig 6 diurnal cycle pattern (March 27, 12 pm to March 28, 12 pm) representative of other days?
L241-249: This is a great idea to investigate the diurnal cycle and the potential meteorological drivers. There not many studies that do this and I think this will contribute greatly to the literature. However, this analysis needs to be more substantial. Please include a description of the HRRR model in section 2. I am interested to know the spatial resolution of the model and if you can resolve the different sites using the model data. Does this help explain the differences in pollen concentration between the 3 sites (in the 1st paragraph of section 3.3).
L245: why are you considering only “ad hoc” meteorological variables? It would be great to have a hypothesis of what meteorological variables you expect to affect the pollen concentration. Please list all the meteorological variables you study.
L246: What r values correspond to “moderately correlated”? I would like to see figures showing the correlation between all the meteorological variables. Variables that aren’t significantly correlated can also be informative. There are a huge number of studies that have looked at the correlation between meteorological variables and pollen concentration, please cite at least a few.
L 246-247: “Overall, pollen count is moderately correlated with weather parameters, with temperature and boundary layer heights having a positive correlation, and relative humidity having a negative correlation.” Can you describe how temperature, boundary layer height and humidity affect pollen concentration?
L267-269: “Several previous studies also compared their automatic pollen monitor with a Hirst type sampler, another NAB certified sampling method.” Please cite studies.
L271-275: Is the point of these sentences to say that Oteros’ BAA500 then a better sensor than the APS300 because it has a higher correlation?
L295: Please check the year of the citation, I think 1970 is not correct.
L297-299: “Fig. 5A indicates that Marietta had the most concentrated and shortest pollen season while the other two sites had an earlier start date of the peak pollen season, multiple peak days, and a longer pollen season.” How did you define the start date and the end date of the pollen season? You need to quantify this so you can compare between the two sites.
L310-311: “Chen 310 et al., 2020; Husam et al., 2018” These aren’t listed in the references.
Reviewer 2 Report
Dear authors/editors,
The presented work/paper is correctly written and clearly explained. Despite that, I leave some suggestions for corrections and improvements:
Lines 184 and 185: “However, the APS-300 also identified Chenopodium, a large group of annual weedy plants, as a 185 dominant pollen species in the peak season, but the Rotorod did not.” The question is: is more likely that APS-300 performed better than Rotorod? Or this is likely a false positive from APS-300?
If both can be true, please specify those or other possible explanations.
Line 113/line 187: line 113 refers to “week 13”, which is not directly related with figure 3 “time nomenclature”. Is it possible that figure 3a) and 3b) can get double “time nomenclature” with week number? Alternatively, to use only one but more convenient “time nomenclature. Take into account what can be easier for readers.
Line 187: In figure 3 b) consider in x axis replacing “Device” by “APS-300 equipment location” or “device location”.
Only figure 4 uses/presents r squared. All tables and other figures use only r. Any reason?
Lines 248 and 249: “The correlation with surface-level wind speed 248 was significant at both Southface and Emory but was not significant at the Marietta site”. Could you speculate a bit (with right references) about the importance of wind? Is it important because of higher emissions? Wind transport? ….
Lines 99 to 107: In general, I consider that the APS-300 equipment should be better defined. I leave some examples of questions that would be of interest if you can detail. In lines 101 to 103 the text refers that:
“The imaged particles are classified into pollen species by machine learning algorithms, and the resulting pollen counts of each species are converted into a daily concentration of pollen granules”
This machine learning algorithms are stable? Can be country/region calibrated? Is this algorithm opensource? or scientifically transparent/valid?, can be calibrated/adjusted by user?
Why the word “daily” above? It can be hourly!? Line 105: “…real-time”. Can you be more specific? can be as low as 1 hour?, 5 minutes?, ?
In line 264: “…identify up to 23 pollen species in near real-time…”. How much time to wait for results? Few seconds, minutes?
Is that algorithm something that the developer can improve in the future and the equipment hardware remain valid?
Line 268: “APS-300 system needs to strengthen the image library of 268 these species to better train the pollen recognition algorithm …” Is this library something that will progress and remain compatible with the equipment?
The processing of images is done in a cpu of the equipment? Or are the images processed remotely in a faster computer? Is there an option to run that algorithm in a more “perfect way” even if it would take longer?
How much time it requires to process one image?
Today, it is available a new model APS-330. What are the differences? Is it mostly the Algorithm?
Reviewer 3 Report
The manuscript is well structured and presented and the device is well described.
Please see attached

Round 2
Reviewer 1 Report
L50: $3.4 billion in medical costs are in the Saha paper, but citation should be Meltzer and Bukstein 2011
L117-118: “Currently, it cannot identify the families of Poaceae and Cupressaceae” Do you mean you cannot identify the genera within the families Poaceae and Cupressaceae ?
Figure2a inset still covering up the APS300 pollen timeseries (dates around mid May)
Figure2B inset still covering up the APS300 pollen timeseries (dates mid Oct – mid Nov)
Fig 4, inset is still covering up pollen timeseries (dates in late Mar -early Apr)
L417-418: “As far as we know, our study is the first attempt to explore the association between hourly pollen concentrations and weather conditions. Our analysis was exploratory in nature and there is no previous research to refer to.” I don’t think you can make this claim. I did a quick google scholar search for “pollen diurnal meteorology” and found a lot of references. Precipitation is very important to modifying pollen concentration (you mention this in L424, Mikhailov et al. 2021) I think it would be great to include precipitation in your analysis.
Fig3 A and B: If you could use the same colors to represent each pollen taxa between the plot A and B, that would make the figures easier to interpret (i.e poaceae is blue in A and pink in B)
Fig 5A I think my previous comment on defining the start and end date of the pollen season was misunderstood. I was hoping to get a definition such as definition described in this paper Jato et al. 2006, https://doi.org/10.1007/s10453-005-9011-x
Table 1: The request to be consistent in naming with either scientific or common names still needs to be addressed
